# Peer J

# Rapid cessation of acute diarrhea using a novel solution of bioactive polyphenols: a randomized trial in Nicaraguan children

Arthur Dover[1], Neema Patel[2] and KT Park[3]

[1] Aptos Travel Clinic, Aptos, CA, USA
[2] LiveLeaf, Inc., San Carlos, CA, USA
[3] Division of Gastroenterology, Hepatology, and Nutrition, Department of Pediatrics, Stanford University School of Medicine, USA

## ABSTRACT

**Goal**. We assessed the effectiveness of bioactive polyphenols contained in solution (LX) to restore normal bowel function in pediatric patients with acute diarrhea.

**Background**. While providing oral rehydration solution (ORS) is standard treatment for diarrhea in developing countries, plant-derived products have been shown to positively affect intestinal function. If a supplement to ORS resolves diarrhea more rapidly than ORS alone, it is an improvement to current care.

**Study**. In a randomized, double-blind, placebo-controlled cross-over study, 61 pediatric patients with uncontrolled diarrhea were randomized to receive either ORS + LX on day 1 and then ORS + water on day 2 (study arm) or ORS + water on day 1 and then ORS + LX on day 2 (control arm). Time to resolution and number of bowel movements were recorded.

**Results.** On day 1, the mean time to diarrhea resolution was 3.1 h (study arm) versus 9.2 h (control arm) ($p = 0.002$). In the study arm, 60% of patients had normal stool at their first bowel movement after consumption of the phenolic redoxigen solution (LX). On day 2, patients in the study arm continued to have normal stool while patients in the control arm achieved normal stool within 24 h after consuming the test solution. Patients in the control arm experienced a reduction in the mean number of bowel movements from day 1 to day 2 after consuming the test solution ($p = 0.0001$). No adverse events were observed.

**Conclusions.** Significant decreases in bowel movement frequency and rapid normalization of stool consistency were observed with consumption of this novel solution.

## INTRODUCTION

Diarrhea is the second leading cause of death in children under the ages of 5 years in developing countries (*Johansson, Wardlaw & Binkin, 2009*), a most concerning statistic as diarrhea may be prevented and treated. Acute diarrhea can lead to severe dehydration and electrolyte imbalance by loss of fluids, electrolytes, and nutrients (*Munos, Fischer Walker & Black, 2010*). Oral rehydration therapy was initially developed to replace cholera-induced fluid loss (*Pierce et al., 1969*; *Sentongo, 2004*), but has expanded to include diarrhea incited

Corresponding author
KT Park, ktpark@stanford.edu

by other pathogens (*Hirschhorn, 1980*; *Nalin et al., 1979*; *Pizarro et al., 1983*). The World Health Organization (WHO) standardized an oral rehydration solution (ORS) containing sodium, potassium, chloride, citrate, and glucose (*Atia & Buchman, 2009*). Although ORS assists in diarrheal management, it does not reduce the duration of diarrhea or fecal volume (*Canai et al., 2007*). Instead, implementing ORS can increase stool volume in children during acute episodes (*Sarker et al., 2001*; *El-Mougi et al., 1994*). In order to optimize efficacy, the WHO recommended a modified ORS with reduced osmolarity, administration of zinc gluconate, non-digestible carbohydrates, rice powder, and probiotic bacteria—all with mixed results (*Gregorio et al., 2007*; *Basu et al., 2007*; *Narayaappa, 2008*; *Hoekstra et al., 2004*; *Passariello et al., 2011*).

In developing countries, attempts for rehydration using readily available household beverages often exacerbate intestinal fluid loss by elevating osmotic load and disrupting water and electrolyte absorption (*Munos, Fischer Walker & Black, 2010*; *Sentongo, 2004*). However, the proper use of ORS and public health measures in Nicaragua including widespread rotavirus vaccinations in infants has been associated with a 35% reduction in childhood mortality over 5 years in the early 1980s. This rate has since remained relatively constant (*Gibbons, Dobie & Krieger, 1994*). Currently, antibiotics serve a very limited role in treating diarrhea in children and the utility of anti-motility agents is either contra-indicated or controversial due to heightened infection risks and adverse effects.

The use of naturopathic medicines in rural or developing populations is often attributed to the inaccessibility of western medicines for common infectious illnesses and a traditional belief in the natural, beneficial properties of plant and plant-derived products. Recent investigations into the efficacy of various plants have identified that their phytochemicals can affect intestinal function and motility (*Njume & Goduka, 2012*; *Bukhari et al., 2013*; *Velázquez et al., 2012*; *Rajan et al., 2012*; *Patil et al., 2012*; *Ezeja et al., 2012*) and provide antibacterial activity (*Abbassi & Hani, 2012*; *Knipping, Garssen & van't Land, 2012*; *Ismail, Sestili & Akhtar, 2012*; *Mariita et al., 2011*; *Assam et al., 2010*). While commercial extraction and processing of these compounds can reduce their viability, a novel processed plant extract composition, LifeDrops (LiveLeaf Inc., San Carlos, CA), captures the bioactive potential of live plant cells. The LifeDrops solution contains a complete complex of green tea (*Camellia sinensis*) and pomegranate *(Punica granatum)* incorporating biologic co-factors key to delivering the full capability of the plants' immune response, termed LiveXtract solution (LX). The mechanism behind LiveXtract solutions is based upon a transient polyphenol reaction common to nearly all higher plants. The site activation of this reaction by the body's enzymes delivers a powerful synergy of localized injury protection, toxin neutralization, and attenuation of inflammation that cannot be produced by conventional. polyphenol extracts (*Romier et al., 2009*; *Vauzour et al., 2010*; *Taylor, Hamilton-Miller & Stapleton, 2005*; *Biasi et al., 2011*; *Romier-Crouzet et al., 2009*; *Kim, Rajalah & Wu, 2008*).

The objective of this study is to compare the efficacy of ORS + LX (LifeDrops) versus ORS + water (placebo) in reducing the incidence and frequency of loose stools and associated gastrointestinal symptoms of pediatric patients with acute diarrhea in

Nicaragua. We hypothesized that the addition of LifeDrops to standard ORS, compared to ORS alone, would reduce the time to normalization of stools and digestive function.

## MATERIALS & METHODS

### Study design

This randomized, double-blinded, placebo-controlled study was conducted at a government-funded community health clinic in Managua, Nicaragua, between August and December 2010. Following torrential rains and flooding in the region from tropical storms Agatha and Matthew, there was a substantial increase in the incidence of consultations for acute diarrhea. With approval of the institutional review board of the Universidad Centroamericana de Ciencias Empresariales (IRB 2010013, registered ISRCTN57765025)), treatment-naïve, previously healthy pediatric patients between 2 and 17 years of age who arrived at the clinic with uncontrolled acute diarrhea within 48 h prior to presentation were enrolled in the study. Written informed consent was obtained from the parents or legal guardians of patients who met the inclusion criteria.

### Statistical analysis

Sample size calculations were based on studies in acute diarrhea using standard ORS treatment in non-cholera pediatric patients. A sample size of $\geq 30$ patients per arm was based upon detecting at least a 15% difference in the duration of diarrhea at the 5% significance level with 80% power. Differences between means of parametric data were analyzed with the Student's $t$-Test, with significance set at 0.05 level. Nonparametric data were analyzed with Chi-squared and Wilcoxon rank-sum tests.

### Study inclusion

All patients who presented to the clinic were assessed and included if they had acute gastroenteritis, including diarrhea, for 48 h or less. Diarrhea was defined as three or more loose or liquid stool per day. Patients were excluded from the study if they had a history of uncontrolled emesis, grossly bloody stool, fever, clinical signs of a coexisting acute systemic illness (e.g., meningitis, sepsis, pneumonia), underlying chronic disease (e.g., heart disease, cystic fibrosis, diabetes), food allergies or other chronic gastrointestinal diseases, admitted use of probiotic agents in the previous 3 weeks or antibiotics or anti-diarrheal medication including over-the-counter and herbal substances in the previous 2 weeks, generalized cachexia, any signs of internal bleeding or drug abuse, or any condition assessed by standard of care to place unnecessary risk if placed on ORS alone. Every patient had a microscopic stool evaluation at the time of enrollment, and those positive for an intestinal protozoan infection were excluded from the study.

#### ORS + LX vs. ORS + water

After study eligibility was determined and consent was obtained, patients were randomized to one of two arms based upon a computer-generated random number listing. The study arm consisted of ORS + LX (LifeDrops) on day 1, then ORS + water on day 2. The control arm consisted of ORS + water on day 1, then ORS + LX on day 2. Patients in

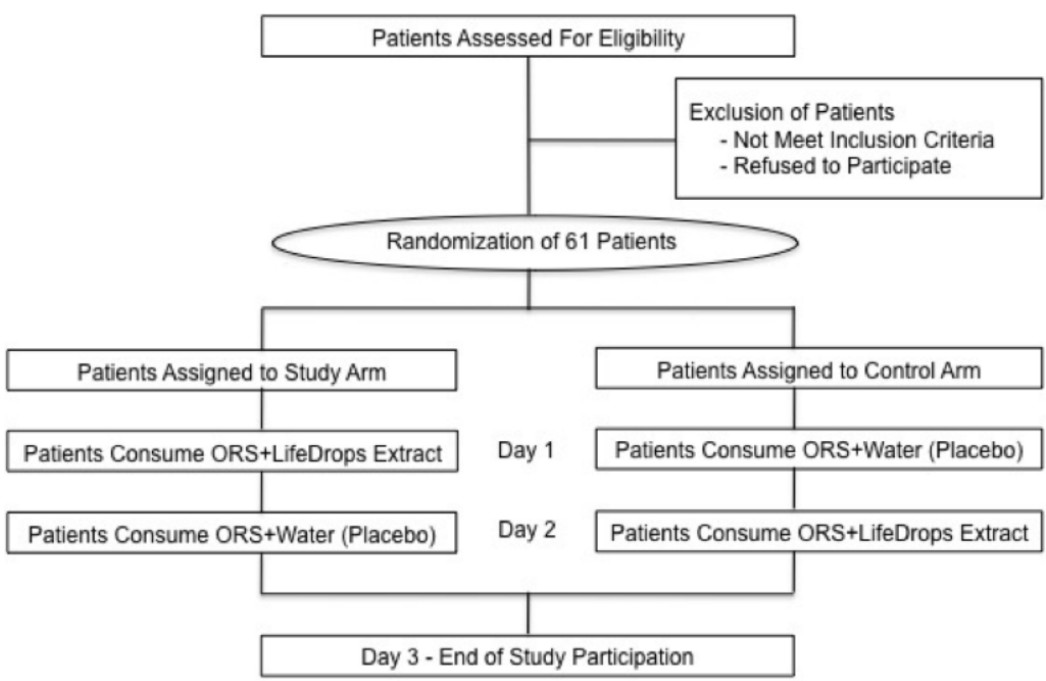

**Figure 1 Study design and patient disposition.** Patients randomized to the Study Arm were given a mixture of oral rehydration salts (ORS) and LiveXtract (LX) solution (test solution) on day 1 and then a mixture of ORS and water (placebo) on day 2. Patients randomized to the Control Arm were given a mixture of ORS and water on day 1 and then a mixture of ORS and LiveXtract solution on day 2.

**Table 1 Serving size of LiveXtract solution administered based upon the weight of the patient.**

| Weight of patient (kg) | Serving size (mL) |
|---|---|
| 10–19 | 3.5 |
| 20–29 | 7.0 |
| 30–39 | 10.5 |
| 40–49 | 14.0 |
| 50–59 | 17.5 |

both arms were given one of the blinded solutions on the first day of clinical evaluation and subsequently monitored by clinic staff for two hours (Fig. 1). While not a true cross-over study design, patients were given the solution on day 2 that was opposite of what was provided on day 1 in order to assess if there were any differences in symptom resolution. A graduated dosing scale, based on patients' weight, determined the volume of LiveXtract solution administered (Table 1). In the control arm, the same volume of water was added to the ORS in order to equal the 25 mL total fluid volume given to patients in the study arm. To enhance the uptake of the test solutions, the ORS contained an added commercial artificial flavor and coloring produced by the Acama company in Central America. Zinc gluconate was not administered during the study period.

**Table 2 Demographics of study population given oral rehydration solution and water (ORS + water) and oral rehydration solution and LiveXtract solution (ORS + LX).**

| Demographics | Study arm (*n* = 30) (ORS + LX) | Control arm (*n* = 31) (ORS + water) | *P* |
|---|---|---|---|
| Age, mean (SD), years | 8 (5.33) | 7 (5.53) | 0.51[a] |
| Weight, mean (SD), kg | 32 (19.89) | 27 (19.32) | 0.31[a] |
| Sex (male/female) | 13/17 | 18/13 | 0.16 (study arm)[b] 0.11 (Control arm)[b] |

**Notes.**

[a] Student's *t*-test, significance set at 0.05.
[b] Chi-squared test, significance set at 0.05.

Two hours after administration of either solution on day 1, the patients were released from the clinic with a maintenance amount of ORS for the next 24 h. All patients were asked to return within 24 h on day 2 for administration of the alternate solution.

## Outcome measures

The primary outcome measure was the time elapsed from the initial ingestion of ORS + LX or ORS + water to any subsequent "unformed" stool, based on the Bristol Stool Scale (BSS), a validated method of visually categorizing stool in 7 appearances based on stool shape and consistency. It has been shown to have reproducibility in pediatric cohorts (*Lane et al., 2011*; *Lewis & Heaton, 1997*). We considered any BSS >4 to be "unformed" and ≤4 to be "formed." The clinical staff ranked the stool during the first 2 h after solution ingestion and parents were trained to score and report the ranking of each bowel movement while away from clinic.

The secondary outcome measures were defecation urgency and bloating/gas following fluid consumption, and a qualitative rating of abdominal pain (for patients able to comprehend and follow directions) on a numeric scale of 0 (none) to 10 (worst imaginable/continual) at 30, 60, 90, and 120 min after consumption of either solution on both day 1 and day 2.

## RESULTS

### Patient demographics

A total of 61 patients were enrolled in this study with 30 patients randomized to the study arm (ORS + LX) and 31 patients to the control arm (ORS + water) on day 1. All subjects were found to be free of protozoan infection by microscopic stool examination, but the specific etiologies of their diarrhea were not definitely known, as per standard of care in this clinical care setting. The patients in each arm were comparable in age (mean age of 8 vs. 7 years, $p = 0.51$) and weight (mean weight of 27 vs. 32 kg, $p = 0.31$), but with more females present in the study arm and more males in the control arm (Table 2).

### Response to solutions consumed on day 1

The summary of results shown in Fig. 2 demonstrates that patients in the study arm achieved a time-to-last unformed stool (a BSS ranking of 4 or less) in a mean elapsed time

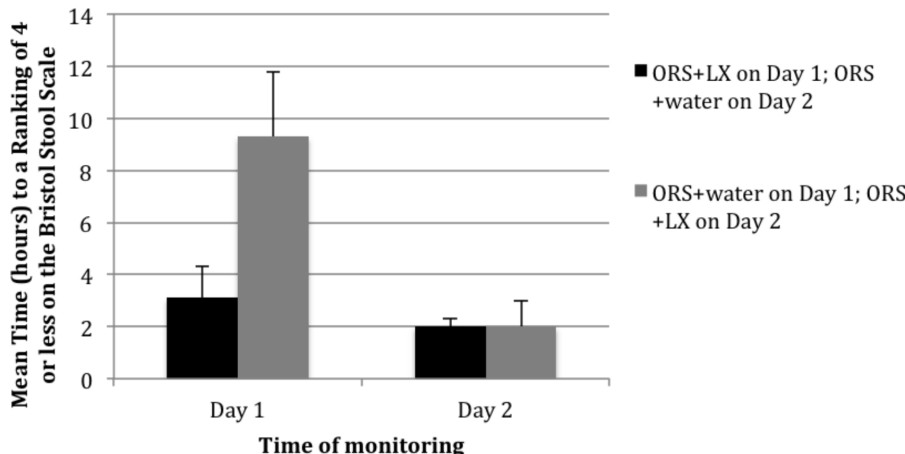

**Figure 2  Mean time (hours) to resolution of acute diarrhea following consumption of either a mixture of oral rehydration salts (ORS) and LiveXtract (LX) solution (test solution) or a mixture of ORS and water (placebo) on day 1 and day 2 of the study.**

of 3.1 h versus 9.3 h among patients in the control arm ($p = 0.002$) on day 1 of the study. In the study arm, 60% of the patients had their first bowel movement with a BSS of 4 or less after consuming the ORS + LX. In the control arm, only 29% of the patients had their first bowel movement with a BSS of 4 or less after ORS + water consumption. At the second movement on day 1, 82% of patients in the study arm versus 35% of patients in the control arm reported stools with a BSS rating of 4 or less ($p < 0.001$).

Patients in the study arm also experienced a longer mean time between bowel movements after solution consumption: 3.7 h in the study arm and 2.8 h in the control arm, which did not achieve statistical significance. The mean time between the first and second bowel movements after consumption was 7 h in the study arm versus 4.4 h in the control group ($p = 0.02$).

### Response to solutions consumed on day 2

When patients returned on day 2 of the study, those in the study arm received ORS + water while those in the control arm received ORS + LX. After 2 h, all patients in the study arm reported stool with a BSS rating of 4 or lower. Patients in the control arm subsequently reported resolution of their diarrhea at a rate comparable to that noted on day 1 for patients in the study arm (Fig. 2). On day 2, patients in control arm had a mean ranking of stool of 4.5 prior to consuming the ORS + LX, which decreased to 3.2 by the first bowel movement after consumption and further decreased 2.2 by the end of day 2 ($p < 0.01$). Patients given ORS + water on day 1 had a mean number of 4 bowel movements that declined to a mean of 2 after receiving ORS + LX on day 2 ($p < 0.01$).

### Secondary outcome measures

Patient-reported responses (e.g., abdominal pain) were incompletely collected during November and December of 2010, resulting in responses from only 10 study arm patients and 7 control arm patients, sample sizes too small for meaningful analyses. The rating of

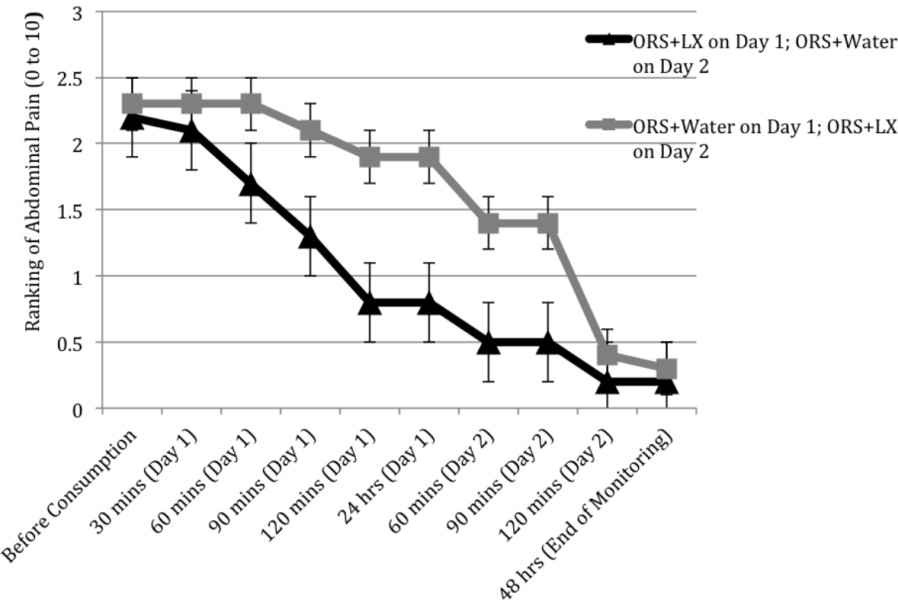

**Figure 3** Mean ranking of abdominal pain over two days at 30 min intervals, after consuming either a mixture of oral rehydration salt (ORS) and LiveXtract (LX) solution (study arm) or ORS mixed with water (control arm).

gas and bloating was comparable between the two arms over the two days, but patients in the control arm did report improvement in their levels of abdominal pain and urgency of defecation soon after consumption of the ORS + LX on day 2 (Figs. 3 and 4). The rating of abdominal pain in patients in the control arm decreased to levels comparable to that reported by patients in the study arm within 2 h after consumption of ORS + LX and was essentially identical to patients in the study arm at the end of the study period (Fig. 3). The rating of defecation urgency, despite remaining unchanged for 24 h after consumption of ORS + water, declined substantially within 60 min post-ORS + LX consumption and continued to decline during the study period (Fig. 4). No adverse events were reported or observed during the study due to ingestion of either of the solutions, and none were reported to the clinic staff after the conclusion of the study period. Additionally, relapse of symptoms was not subsequently reported to the clinic staff.

## DISCUSSION

In this randomized controlled trial, we demonstrate that compared to ORS alone, supplementation of a novel LiveXtract solution (LifeDrops, San Carlos, California, USA) significantly decreased resolution time of acute diarrhea and accelerated normalization of stool consistency. All patients in the study experienced faster resolution of their diarrhea after receiving ORS + LX, and all soon achieved normalization of stool consistency. The intervention cohort receiving ORS + LX had normalization to BSS $\leq 4$ stool consistency and frequency by the end of day 1. Similarly, control patients who received ORS + LX on day 2 (after receiving ORS + water on day 1) reported comparable efficacy by the end of day 2.

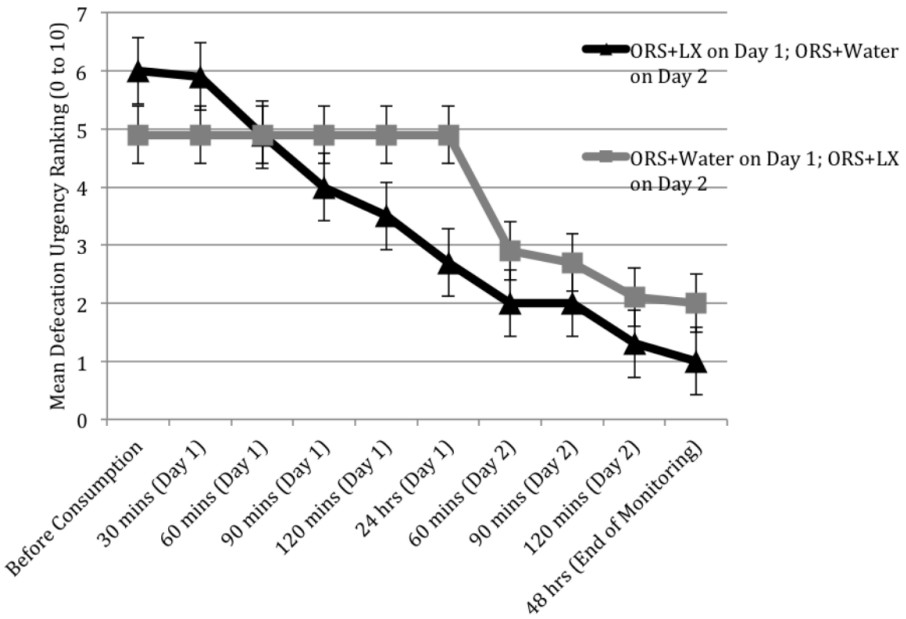

**Figure 4** **Mean ranking of urgency to defecate over two days at 30 min intervals, after consuming either a mixture of oral rehydration salt (ORS) and LiveXtract (LX) solution (study arm) or ORS mixed with water (control arm).**

Secondary outcome measures of abdominal pain and defecation urgency also improved for both cohorts upon initiation of ORS + LX by the end of the same day. By the end of the monitoring period on day 2, patients in the control cohort noted a reduction in both adverse symptoms similar to patients in the intervention cohort reported by the end of monitoring on day 1. No adverse events were reported or observed in any patient receiving ORS + LX.

One limitation of our study is the lack of infectious pathogen identification in subjects' acute diarrheal illness. This study was conducted at a government-funded community health clinic in Managua, Nicaragua following torrential rains and flooding in this region in late 2010. Resource limitations and prioritization of streamlined humanitarian efforts made pathogen identification difficult in the context of a clinical trial, although subjects with evidence of any protozoa by light microscopy were excluded and referred for treatment. Given our hypothesis that the LiveXtract solution maintained the natural antibacterial properties of *Camellia sinensis* and *Punica granatum* within the enteric tract after consumption, we theorize that plant extracts rich in polyphenols have the potential to stimulate innate host immune processes by action of phyto chemicals from natural plant immunity and to antagonize common enteric pathogens responsible for acute bacterial and viral gastroenteritis. Previous literature has identified waterborne enteric pathogens as likely gram negative bacterial species, such enterotoxigenic *Aeromonas, Campylobacter, Salmonella, Shigella,* and enterotoxigenic *Escherichia coli*, which all thrive in warm freshwater environments (*Burke et al., 1983*; *Ashbolt, 2004*), reproduced in the natural elements present in our study.

Another limitation is that our data represent a snapshot of a narrow study timeframe and one specific geographical location. While acknowledging the weaknesses of our study, we also recognize the strength of our study's randomized study design. For a prospective pilot study, we surpassed adequate enrollment numbers to show a clear statistical difference between ORS + LX vs. ORS + water. Among the individual subjects, we showed distinct reproducibility of the treatment effect upon introduction of ORS + LX between individual patients.

The limited number of pediatric patients who provided data for secondary outcome measures did not permit statistical analyses of the changes in these patients' quality of life. However, the data do show a trend of reducing abdominal pain and defecation urgency with consumption of the polyphenol supplement, which needs to be verified in future clinical outcome studies.

Preventing and reducing morbidity and mortality from acute diarrheal illnesses causing dehydration is a significant public health concern, and remains an on-going global health initiative. Although the use of ORS to restore intravascular fluid losses remains the standard of care in most clinical scenarios, there are limited clinical alternatives aimed to actively shorten the time of acute diarrheal fluid and electrolyte losses. LiveLeaf LifeDrops solution potentially represents a novel approach to effectively reduce morbidity and mortality from acute diarrhea illnesses in certain situations. In this preliminary study, we report the results of the first prospective clinical trial using this unique supplement to ORS. Published literature in this area includes several negative studies of the addition of rice or non-digestible carbohydrates to ORS (*Sarker et al., 2001*; *El-Mougi et al., 1994*; *Hoekstra et al., 2004*; *Faruque et al., 1997*; *Khan et al., 2005*). Further literature review of the efficacy of trace elements such as zinc (*Gregorio et al., 2007*; *CHOICE Study Group, 2001*) and probiotics (*Basu et al., 2007*; *Wadhwa et al., 2011*) to reduce acute diarrheal disease burden are mixed. The current recommendation of the World Health Organization (WHO) is to provide low osmolarity ORS and zinc supplementation for 10 to 14 days (*Burke et al., 1983*), which is associated with reduced time to resolution of diarrhea in several clinical studies (*El-Mougi et al., 1994*; *Gregorio et al., 2007*; *Boran et al., 2006*; *Dutta et al., 2000*; *Patel, Badhoniya & Dibley, 2013*), but with times substantially longer than the 3 h noted in patients given the LiveXtract solution.

Future directions should be aimed at understanding the mechanisms of phytochemicals as potential consumable agents effective in acute infectious gastroenteritis. Elucidation of the molecular basis of the phytochemicals' action on enteric pathogens—through a detailed biochemical pathway—should be pursued, as well as their possible interaction with innate host intestinal immune systems, supported by microbiota analysis. Clinical research efforts should also be directed to test the robustness of our initial efficacy data through reproducibility while subject to contextual study variability.

## CONCLUSION

In this randomized clinical study, pediatric patients with acute diarrhea experienced rapid improvement of stool consistency following ingestion of the LiveXtract solution. Further

clinical data are necessary in order to corroborate these results, but the rapid resolution in pediatric patients in this study suggests a well-tolerated, safe, and effective option for the resolution of acute diarrhea syndrome.

## ACKNOWLEDGEMENTS

We thank Dr. Telma Noguera (Instituto Centroamericano de Investigación Clínica, Managua, Nicaragua) and Mr. Rob Wotring (LiveLeaf Inc.) for their contribution in coordination of data collection.

### Funding

All aspects of the study were funded by LiveLeaf, Inc. The funders had no role in study design, data collection and analysis, decision to publish, or preparation of the manuscript.

### Grant Disclosures

The following grant information was disclosed by the authors:
LiveLeaf Inc.

### Competing Interests

Dr. Arthur Dover and Dr. KT Park serve as scientific advisors for LiveLeaf, Inc. Neema Patel served as a consultant to LiveLeaf, Inc.

### Author Contributions

- Arthur Dover conceived and designed the experiments, performed the experiments, wrote the paper, reviewed drafts of the paper.
- Neema Patel analyzed the data, wrote the paper, prepared figures and/or tables, reviewed drafts of the paper.
- KT Park conceived and designed the experiments, analyzed the data, wrote the paper, reviewed drafts of the paper.

### Human Ethics

The following information was supplied relating to ethical approvals (i.e., approving body and any reference numbers):

Ethics Committee of Universidad Centroamericana de Ciencias Empresariales; approval number 2010013.

### Clinical Trial Ethics

The following information was supplied relating to ethical approvals (i.e., approving body and any reference numbers):

Ethics Committee of Universidad Centroamericana de Ciencias Empresariales, Clinical trials registration number for this study:

ISRCTN57765025.

## Clinical Trial Registration

The following information was supplied regarding Clinical Trial registration:

Ethics Committee of Universidad Centroamericana de Ciencias Empresariales, Clinical trials registration number for this study:

ISRCTN57765025.

## Supplemental Information

Supplemental information for this article can be found online at http://dx.doi.org/10.7717/peerj.969#supplemental-information.

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
