# Peer review of "Rapid cessation of acute diarrhea using a novel solution of bioactive polyphenols: a randomized trial in Nicaraguan children"

_PeerJ, doi:10.7717/peerj.969_

## Round 0.1 · original submission · Minor Revisions

The paper can be accepted pending revisions suggested by the reviewers. In addition, raw data should be uploaded for the next review. This allows verification and assessment of statistics. I look forward to the revised version.

·

Basic reporting

What is the affiliation no: 4? I can see 1, 2, 3 but not 4.

Experimental design

No comments

Validity of the findings

No comment

Additional comments

Dear Editor,
This is a randomized, double- blinded, placebo controlled cross-over study performed in Managua, Nicaragua. The study shows that adding the solution LX that contains green tea and pomegranate to ORS shortens the time to diarrhea resolution. This complex solution may help children have relief at an earlier time and also improve the quality of life in a shorter time as it looks to reduce patient reported symptoms such as abdominal pain.

Some questions to be answered.
1. What is the affiliation no: 4? I can see 1, 2, 3 but not 4.
2. Results section: The authors mentioned that the mean weight was comparable. Can they give p value for mean weight of 27 vs. 32 kg.
3. Figure 2: The authors can put * and give p where significant on graph
4. Results section - Secondary outcome measure: Only 10 in study arm and 7 in the control arm provided data for patient-reported responses like abdominal pain. These are pretty low numbers. This needs to be mentioned in discussion as a limitation. It looks like there may be significant difference in mean rating for abdominal pain between study vs. control group at 120 min. after ingestion of test dose. However the number of patients are small so it may not be logical to give a p value. It might be nice difference if they had higher number of patients that provided these data.
Figure 1: There are two different legends for figures. Do not need the statement given below as the flowchart explains the statement. “Patients randomized to the Study Arm were given a mixture of oral rehydration salts (ORS) and LiveXtract (LX) solution (test solution) on day 1 and then a mixture of ORS and water (placebo) on day 2. Patients randomized to the Control Arm were given a mixture of ORS and water on day 1 and then a mixture of ORS and LiveXtract solution on day 2.”
5. Figure 2: The statement is already in the text so don’t need here. “Resolution of diarrhea was based on stool have a ranking of 4 or less on the Bristol Stool Scale (BSS). Note that by the end of day 2, those given the placebo on day 1 but then given the LiveXtract solution on day 2 (the cross-over step) achieved BSS rankings consistent with resolution of diarrhea in a mean time comparable to that of those who had consumed the LiveXtract solution on day 1.
6. Figure 5 is given in discussion section. Better give in results section although not sure if it’s a good idea to compare different studies directly in terms of “mean duration of diarrhea” as the designs are different.

·

Basic reporting

Overall the reporting is good.
1.Trial entry criteria was not properly stated. Whether all patients were considered? The criteria for acute diarrhea was not clearly stated
2.Definition of diarrhea in this study was also not clearly stated, although as outcome measure the author used stool consistency based on the BSS
3. The method of randomization should be detail out further

Experimental design

This is a randomised trial- a good study design, but using a cross-over design might not be appropriate in a disease like acute gastroenteritis. However this was also highlighted by the authors.

Validity of the findings

The calculation of sample size need further clarification. The use of 5% change in duration, in my opinion is too low a value, the author should give a reference, to which this figure is based.
using self reporting by patient on the consistency of the stool is subjected to reliability issue. although the patients were trained. It is better to show that the self reporting by the patient was valid prior to study commencement.

Additional comments

Generally the study is acceptable.

·

Basic reporting

This is a well written article. The literature of the subject has been reviewed. The method is sound and well designed. The findings are discussed well.

Experimental design

No major flaws in the design

Validity of the findings

As the methods are very strong the findings are valid. I am surprised the way that this chemical stops diarrhoea.

Additional comments

Well done

---

## Round 0.2 · Major Revisions

Although the revised version is satisfactory to two of the earlier reviewers, Reviewer 4 (who was asked to comment on the statistics and Clinical Trial design) is particularly concerned about the design and data analyses. The authors are advised to look through the comments of Reviewer 4 (and their attached document) very carefully.

·

Basic reporting

No comments

Experimental design

No comments

Validity of the findings

No comments

Additional comments

The authors have provided appropriate responses and did the required orrections in the manuscript. It can be published in this way.

·

Basic reporting

no comment

Experimental design

no comment

Validity of the findings

no comment

Additional comments

the author has done the correction and given the explanation where appropriate.

Reviewer 4 ·

Basic reporting

The basic reporting is solid.

Experimental design

The experimental design used in this paper is not appropriate. The crossover design should not be used when there is carry-over effect and when the treatment cure the conditions. Both of these two effects are apparent in this paper. Furthermore, the washout period is not specified.

In addition, the statistical methods used of the data analyses are not appropriate.

Validity of the findings

The founding is not valid due to the inappropriateness in the study design and statistical analyses. Please consult a biostatistician in order to rescue the paper.

Additional comments

The paper is innovative and addresses an urgent need. However, it is undermined by the flaws in the study design and in the statistical analyses.

Annotated reviews are not available for download in order to protect the identity of reviewers who chose to remain anonymous.

---

## Round 0.3 · accepted · Accept

All reviews are now in, and the revision is clearly a major improvement. I am pleased to accept the paper in its current form and congratulations.

Reviewer 4 ·

Basic reporting

The paper is solid.

Experimental design

After edition, the study design is appropriate.

Validity of the findings

The study was well conducted and the final analyses were appropriate, thus the study findings are reliable.

Additional comments

I appreciate that the authors addressed my questions very carefully and with great effect. Many thanks for tolerating my tough questions.